

# Dose-dependent and strain-dependent anti-obesity effects of *Lactobacillus sakei* in a diet induced obese murine model

Yosep Ji[1,*], Young Mee Chung[2,*], Soyoung Park[1], Dahye Jeong[2], Bongjoon Kim[2] and Wilhelm Heinrich Holzapfel[1]

[1] Department of Advanced Green Energy and Environment, Handong Global University, Pohang, Gyungbuk, South Korea
[2] Beneficial Microbes Center, CJ Foods R&D, CJ CheilJedang Corporation, Suwon, Gyeonggi, South Korea
* These authors contributed equally to this work.

Corresponding author
Wilhelm Heinrich Holzapfel,
wilhelm@woodapple.net

## ABSTRACT

**Background:** Overweight and abdominal obesity, in addition to medical conditions such as high blood pressure, high blood sugar and triglyceride levels, are typical risk factors associated with metabolic syndrome. Yet, considering the complexity of factors and underlying mechanisms leading to these inflammatory conditions, a deeper understanding of this area is still lacking. Some probiotics have a reputation of a relatively-long history of safe use, and an increasing number of studies are confirming benefits including anti-obesity effects when administered in adequate amounts. Recent reports demonstrate that probiotic functions may widely differ with reference to either intra-species or inter-species related data. Such differences do not necessarily reflect or explain strain-specific functions of a probiotic, and thus require further assessment at the intra-species level. Various anti-obesity clinical trials with probiotics have shown discrepant results and require additional consolidated studies in order to clarify the correct dose of application for reliable and constant efficacy over a long period.

**Methods:** Three different strains of *Lactobacillus sakei* were administered in a high-fat diet induced obese murine model using three different doses, $1 \times 10^{10}$, $1 \times 10^9$ and $1 \times 10^8$ CFUs, respectively, per day. Changes in body and organ weight were monitored, and serum chemistry analysis was performed for monitoring obesity associated biomarkers.

**Results:** Only one strain of *L. sakei* (CJLS03) induced a dose-dependent anti-obesity effect, while no correlation with either dose or body or adipose tissue weight loss could be detected for the other two *L. sakei* strains (L338 and L446). The body weight reduction primarily correlated with adipose tissue and obesity-associated serum biomarkers such as triglycerides and aspartate transaminase.

**Discussion:** This study shows intraspecies diversity of *L. sakei* and suggests that anti-obesity effects of probiotics may vary in a strain- and dose-specific manner.

## INTRODUCTION

Overweight and obesity result from abnormal adipose deposition and function and are considered as major pathophysiological symptoms of metabolic syndrome (*Olufadi & Byrne, 2008*). Originating from insulin resistance, metabolic syndrome may be reflected by several clinical manifestations such as atherosclerosis, hyperglycemia, dyslipidemia, hypertension, reduced high density lipoprotein (HDL) cholesterol and type 2 diabetes mellitus (*Furukawa et al., 2017*). Based on typical pathological symptoms, broadly defined as excessive fat mass in the body (specifically the abdomen), the prevalence of obesity has rapidly increased during the last two decades (*Kobyliak et al., 2017*). Also referred to as "obesity pathogenesis," obesity is considered as a disorder of the energy homeostasis system rather than the result of passive weight accumulation (*Schwartz et al., 2017*). In spite of the recent intensive research input, a deeper understanding of pathogenesis and the underlying mechanisms of obesity are still lacking, while, in fact, the causality of obesity has been explained from different viewpoints and disciplines of science such as genetics, endocrinology and psychology (*Schwartz et al., 2017*).

Following up on classical approaches, recent studies show that the microbiota can play a key role in host obesity and metabolic syndrome (*Gérard, 2016*). Thereby, new clinical diagnostic perspectives were opened on the influence of the gut microbiota on the status of metabolic disorders. This potential has been highlighted in a review by *Boulange et al. (2016)*, at the same time underlining the complex etiology of these disorders. The current understanding of the mechanisms linking the gut microbiota with metabolic syndrome still appears to be "vague" (*Chattophadyay & Mathili, 2018*). Indeed, numerous studies have reported on qualitative and quantitative discrepancies in the microbiota of the gastrointestinal tract (GIT) when comparing healthy subjects with people suffering from metabolic diseases (*Turnbaugh et al., 2006*, *2008*; *Ley et al., 2005*; *Cani & Delzenne, 2009*; *Armougom et al., 2009*).

The International Scientific Association for Probiotics and Prebiotics, after a grammatic correction, has condoned the FAO/WHO consensus definition of probiotics as "live microorganisms that, when administered in adequate amounts, confer a health benefit on the host" (*Hill et al., 2014*). There is general agreement that probiotics support the balance of the host gut microbiota, and scientific evidence is steadily accumulating regarding the positive impact of probiotics on human health such as improvement of immune disorders, inflammatory bowel disease, type 2 diabetes and atherosclerosis (*Amar et al., 2011*; *Kim et al., 2016*; *Ritze et al., 2014*; *Schroeder et al., 2018*; *Vemuri, Gundamaraju & Eri, 2017*). In spite of increasing evidences of beneficial effects, information is still sparse on the way in which gut microbiota communicates with distant sites in the host, and also on the mechanisms underlying their influence on host physiology with regard to (e.g.,) the respiratory system, the skin, brain, heart and host metabolism (*Reid et al., 2017*). The best recognized mechanisms among the studied probiotics appear to be related to colonization resistance, acid and short-chain fatty acid production, regulation of intestinal transit, normalization of perturbed microbiota, increasing turnover of enterocytes, and competitive exclusion of pathogens (*Hill et al., 2014*). Using a high-calorie induced

obesity BALB/c mouse model a single strain of *Lactobacillus casei* IMV B-7280, and a combination of *Bifidobacterium animalis* VKL, *B. animalis* VKB and *L. casei* IMV B-7280 were shown to be effective in reducing weight gain and cholesterol levels, in the restoration of liver morphology and in modulating the gut microbiome in a beneficial manner (*Bubnov et al., 2017*). However, key issues such as strain-specificity and characterization of dose-dependent effects still remain to be solved. For this purpose, the further development of both in vitro and in vivo models appears to be strongly justified. Evidence-based recommendations for probiotics presently suggest a dose of $10^9$ CFU/day or higher (*WGO, 2017*). A former study involving volunteers demonstrated a dose of $10^{11}$ CFU/day (of probiotic strains *B. animalis* subsp. *lactis* BB-12 and *L. paracasei* subsp. *paracasei* CRL-341) to be effective (*Larsen et al., 2006*). For the clinical success of anti-obesity treatment, selection of an optimal dose and an optimal administration time frame of probiotics are considered to be essential for inducing beneficial changes, both in gut microbiome diversity and in the metabolism of obese humans (*Bubnov et al., 2017*).

Various modes of probiotic action were elucidated by using in vitro studies (including development of dedicated in vitro models) while efficacy was investigated by both in vivo (preclinical) studies (*Park et al., 2016*; *Wang et al., 2015*) and clinical trials (*Kadooka et al., 2010*; *Woodard et al., 2009*). These therapeutic benefits were all related to anti-obesity effects of probiotics (*Kadooka et al., 2010*; *Park et al., 2016*; *Wang et al., 2015*; *Woodard et al., 2009*). Yet, the anti-obesity efficacy of probiotics has not been fully elucidated in spite of various clinical trials, and scientific evidence for a "minimal dose effect level" remains relatively sparse (*Tanentsapf, Heitmann & Adegboye, 2011*; *Raoult, 2009*; *Mekkes et al., 2013*). The concept of a minimal effective dose is complicated due to the large (and diverse) number of microbial and host-related factors (*Salminen et al., 1998*), and will also depend on the kind of key criteria and the "end-points" selected. The dose of intolerance is generally considered to be high; thus, allowing a relatively broad "therapeutic window" (*Collins, Thornton & Sullivan, 1998*), it may be difficult to find a suitably low effective dose above the minimal level. Yet, precisely defining an effective dose has remained an arbitrary issue, and thus the pragmatic suggestion by an FAO/WHO Working Group (FAO/WHO, 2002) that "the suggested serving size must deliver the effective dose of probiotics related to the health claim." Convincingly delivering this kind of evidence has remained difficult until this day, in particular for commercial distribution of (food or pharmaceutical) strains claimed to be probiotics. In an early report *Perdigón, Alvarez & De Ruiz Holgado (1991)* suggested a dose related impact of *L. casei* on the secretory immune response and protective capacity in intestinal infections. A placebo-controlled study designed to evaluate the therapeutic value of four different non-antibiotic preparations (including *Saccharomyces boulardii*, and heat-killed microbial strains) indicated a non-significant dose dependency for either prophylaxis or treatment of traveller's diarrhoea (*Kollaritsch et al., 1989*, *1993*). Yet, substantial evidence supports the principle of dose-dependency of probiotics to modulate systemic and mucosal immune function, improve intestinal barrier function, alter gut microbiota, and exert metabolic effects on the host, also in a strain-dependent manner (*Alemka et al., 2010*; *Madsen, 2012*; *Larsen et al., 2013*).

*Everard et al. (2011)* reported a dose-dependent immunomodulation of human dendritic cells by the probiotic *L. rhamnosus* Lcr35, leading, at high doses, to the semi-maturation of the cells and to a strong pro-inflammatory effect. Against this background, the present study was designed with the challenge of involving a hitherto rarely reported species (*L. sakei*) and its potential for alleviation of obesity (in a diet-induced obese (DIO) mouse model). In addition, there was the prospect of gaining additional insights in intra-species (strain-specific) functional diversity by using established biomarkers.

In this study we administered three different 10-fold dose levels of three different *L. sakei* strains separately to a DIO C57BL/6 murine model and monitored body weight during the full experimental period. Organ weights and serum biomarkers were monitored to elucidate the dose-dependent anti-obesity effect of three different *L. sakei* strains.

## MATERIALS AND METHODS

### Animal studies

The animal study was approved by the Ethical Committee of KPC Ltd. in Korea (P150067), in full compliance with ethical standards as specified by Korean law. A total of 5 week-old, specific pathogen free male C57BL/6 mice were supplied from Orient Bio, Korea. Either a high-fat diet (HFD) (Research Diets D12492) (60% kcal fat), or low-fat diet (LFD) (Purina Laboratory Rodent Diet 38057) (12% kcal fat) (negative control) and autoclaved tap water were provided ad libitum, while the animals were housed at 23 °C, 55 ± 10% humidity, in a 12 h light/dark cycle. At the age of 5 weeks mice were fed with either a low-fat control diet containing 12% kcal of total energy from fat (12.41% kcal fat, 24.52% protein, 63.07% kcal carbohydrate (Purina Laboratory Rodent Diet 38057; Purina Korea Inc., Seoul, Korea)) or a HFD with 60% kcal fat ((90% of the fat from lard, 10% from soybean oil), 20% kcal protein, 20% kcal carbohydrate (D12492; Research Diets Inc., New Brunswick, NU, USA)) for 6 weeks. For this study, a HFD of 60% kcal fat was chosen, as this is one of the most commonly used diets to induce obesity and ectopic lipid storage in in vivo studies. Detailed analytical information on the diet composition is given in Table S1 (see also Table 1). The NIH guidelines were followed by providing sufficient cage surface area based on the weight of the mice. In total 120 mice were separated into 12 different groups (five animals per cage and two cages per group) with each group receiving a different treatment. Study design is given in Table 2 and information on the diets in Table 1.

The experiment comprised 1 week of adaptation followed by 6 weeks of obesity induction using a HFD while the LFD group was maintained on LFD feeding. A total number of 110 mice received the test substances, with exception of those with the upper and lower body weights after the 6-week period of obesity induction. All treatments were by oral gavage and were performed twice a day, at the same daytime (10.00 and 17.00), for 7 weeks. Each group was treated with either the microbial culture suspended in phosphate buffered saline (PBS), orlistat suspended in PBS, as chemical control, or only PBS as negative control. Orlistat was provided as Xenical (with 120 mg/g of orlistat as active pharmaceutical ingredient, and microcrystalline cellulose, sodium starch glycolate, sodium lauryl sulfate, povidone and talc as inactive ingredients). The contents of the

Table 1 Diet composition of the low-fat (LFD) and high-fat (HFD) diets used in this study.

**A.**

| Calories (%) | | Ingredients | | | | |
|---|---|---|---|---|---|---|
| | | Protein (%) | Fat (%) | Fiber (%) | Minerals (%) | Vitamins (%) |
| Fat | 12.41% | Arginine (1.26) | Linoleic Acid (1.10) | Crude fiber | Ash (7.25) | Vitamins A, D3, E, K, |
| Carbohydrate | 63.07% | Glycine (0.87) | Linolenic Acid (0.12) | | Calcium (1.20) | Riboflavin, Niacin |
| Protein | 24.52% | Isoleucine (0.82) | ArachidonicAcid (0.02) | | Phosphorus (0.62) | Others |
| | | Leucine (1.47) | Omega-3 Fatty Acids | | Potassium (0.82) | |
| | | Lysine (1.01) | (1.11) | | Others | |
| | | Phenylalanine (0.98) | | | | |
| | | Valine (0.91) | | | | |
| | | Others | | | | |
| Total | 100% | 20 | 4.5 | 3.7 | | |

**B.**

| Calories (kcal%) | | Ingredients (g) |
|---|---|---|
| Fat | 60.00% | Casein, 80 Mesh (200) |
| Carbohydrate | 20.00% | L-Cystine (3) |
| Protein | 20.00% | Maltodextrin 10 (125) |
| | | Sucrose (68.8) |
| | | Cellulose, BW 200 (50) |
| | | Soybean Oil (25) |
| | | Lard (245) |
| | | Mineral Mix, S10026 (10) |
| | | DiCalcium Phosphate (13) |
| | | Calcium Carbonate (5.5) |
| | | Potassium Citrate.1$H_2O$ (16.5) |
| | | Vitamin Mix, V10001 (10) |
| | | Choline Bitartrate (2) |
| | | FD&Blue Dye #1 (0.05) |
| Total | 100% | 773.85 |

Note:
(A) Low-fat diet (Purina Laboratory Rodent Diet 38057); (B) high-fat diet (Research Diets D12492).

Xenical capsules were added to PBS, as explained in Table 1. As orlistat is insoluble in water, it was suspended by vortexing and sonication and then orally administered to the animals. For oral administration each microbial strain was washed twice with PBS and the supernatant discarded after centrifugation. The microbial pellet was resuspended in PBS to suit the dose for administration. On the last day of the experiment, the mice were sacrificed by dislocation of the cervical vertebrata. The organs (liver, femoral muscle, brown adipose tissue, epididymal adipose tissue, subcutaneous adipose tissue and mesenteric adipose tissue) were collected, weighed, and stored at −80 °C. Each perfused liver was embedded in paraffin and sectioned (four μm) on a microtome. Hematoxylin and eosin (H&E) staining was performed on each high dose *L. sakei* group and assessed by light microscopy (Olympus MVX10 microscope, equipped with a DC71 camera; Center Valley, PA, USA; Olympus, Tokyo, Japan).

Serum triglycerides (TG), glucose (GLU), total cholesterol (TC), HDL, low-density lipoprotein (LDL) and aspartate transaminase (AST; a marker of liver toxic injuries of

**Table 2 Study design and animal treatments based on a high-fat (HFD) and low-fat diet (LFD).**

| Group | Feed type | Treatment |
|---|---|---|
| LFD | LFD | 300 μL PBS (non-obese control) |
| HFD | HFD | 300 μL PBS (obese control) |
| Orlistat | HFD | 40 mg/kg suspended in 300 μl PBS |
| CJB38 L | HFD | $1 \times 10^8$ CFU/day of *L. sakei* L338 suspended in 300 μL PBS |
| CJB38 M | HFD | $1 \times 10^9$ CFU/day of *L. sakei* L338 suspended in 300 μL PBS |
| CJB38 H | HFD | $1 \times 10^{10}$ CFU/day of *L. sakei* L338 suspended in 300 μL PBS |
| CJB46 L | HFD | $1 \times 10^8$ CFU/day of *L. sakei* L446 suspended in 300 μL PBS |
| CJB46 M | HFD | $1 \times 10^9$ CFU/day of *L. sakei* L446 suspended in 300 μL PBS |
| CJB46 H | HFD | $1 \times 10^{10}$ CFU/day of *L. sakei* L446 suspended in 300 μL PBS |
| CJLS03 L | HFD | $1 \times 10^8$ CFU/day of *L. sakei* LS03 suspended in 300 μL PBS |
| CJLS03 M | HFD | $1 \times 10^9$ CFU/day of *L. sakei* LS03 suspended in 300 μL PBS |
| CJLS03 H | HFD | $1 \times 10^{10}$ CFU/day of *L. sakei* LS03 suspended in 300 μL PBS |

**Note:**
LFD, low-fat diet (negative control); HFD, high-fat diet; CJB38, CJB46 and CJLS03 denote the three *Lactobacillus sakei* strains; the three dose levels of each strain administered together with the HFD were $1 \times 10^{10}$ CFU/ml (high-dose, H), $1 \times 10^9$ (medium-dose, M) and $1 \times 10^8$ CFU/mL (low-dose, L).

hepatocytes (*Aulbach & Amuzie, 2017*)), were measured using an automated biochemical analyzer BS-200 (Mindray, China) in Pohang Technopark, Pohang (South Korea).

## Microorganisms

*Lactobacillus sakei* strain CJLS03 was isolated from kimchi, while *L. sakei* strains CJB38 and CJB46 originated from human fecal samples. These strains were selected among nine different strains (comprising four *L. brevis*, three *L. sakei*, one *L. plantarum* and one *Bifidobacterium longum*) on the basis of the lowest weight gain in a preliminary study using a DIO mouse model (data shown in Fig. S1).

The three *L. sakei* strains were grown daily in MRS broth (Difco Laboratories INC., Franklin Lakes, NJ, USA) for feeding during the 7-week period of intervention. Strains were grown for 8 h to reach their late log phase and were collected by centrifugation (3,546*g*, 5 min, 5 °C) (Hanil Science Inc., Gangneung, South Korea) and washed two times with PBS. Each strain was prepared in an approximate number of $1 \times 10^{10}$ CFU/ml using a mathematical equation derived from a pre-optimised standard curve (Fig. S2) using optical density by SPECTROstar Nano (BMG Labtech, Durham, NC, USA). A stock suspension of $1 \times 10^{10}$ CFU/mL (high-dose, H) was prepared of each strain, then diluted 10-fold to $1 \times 10^9$ (medium-dose, M) and $1 \times 10^8$ CFU/mL (low-dose, L), respectively, and finally suspended in 300 μl of PBS to be administered to each mouse by oral gavage.

Experimental determinants were statistically calculated using ANOVA and Dunnett's multiple comparison test to distinguish the level of significance based on probability of 0.05 (*), 0.01 (**) and 0.001 (***).

## RESULTS

High-fat diet feeding resulted in a strong increase in body mass as compared to those animals receiving LFD administration (Fig. 1A) over the 48-day feeding period. Moreover,

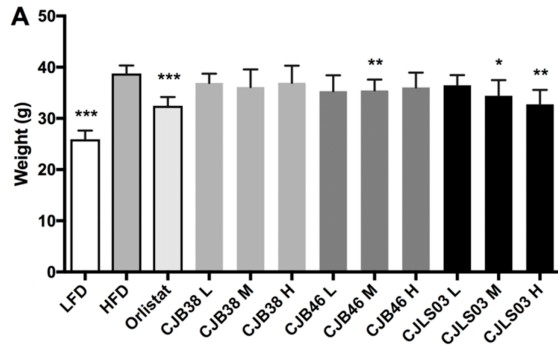

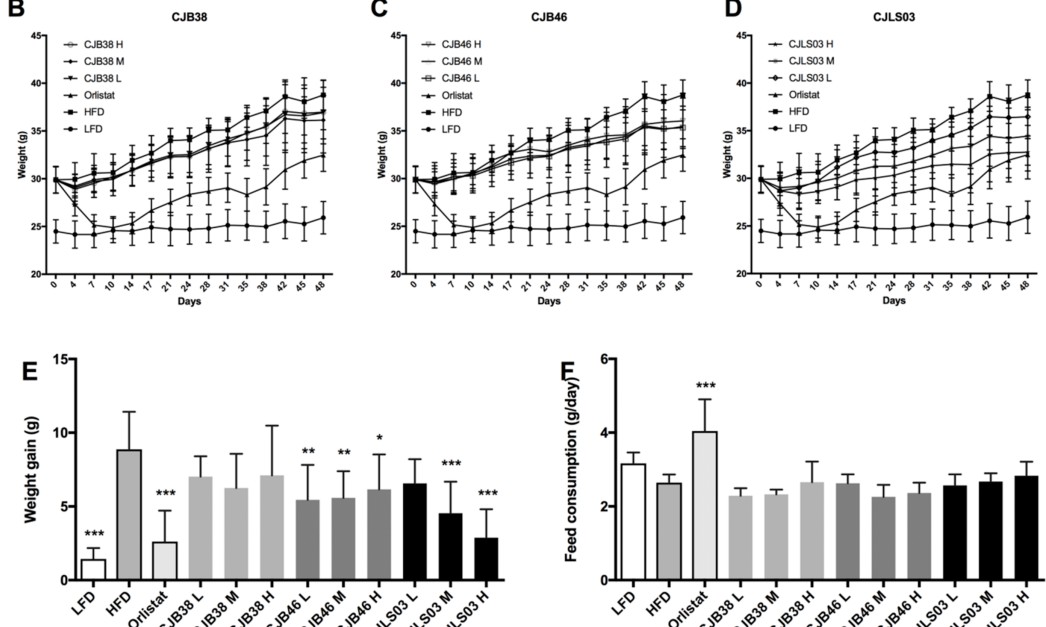

**Figure 1 (A) Body weight after 48 days, and (B–D) increase over the 48-day period; (E) body weight gain after 48 days, and (F) daily feed consumption of each group.** LFD, low-fat diet; HFD, high-fat diet; CJB38, CJB46 and CJLS03 denote the three *L. sakei* strains; the three dose levels of each strain administered together with the HFD were $1 \times 10^{10}$ CFU/mL (high-dose, H), $1 \times 10^9$ (medium-dose, M) and $1 \times 10^8$ CFU/mL (low-dose, L). The values for each index are expressed as the mean ± SD ($n = 10$). Asterisks denote the level of significance compared to HFD as $^*p < 0.05$, $^{**}p < 0.01$ and $^{***}p < 0.001$.

elevated levels of serum biomarkers such as TG, TC, GLU, LDL and AST were detected in the HFD group (Fig. 2), concomitantly with quantitative increases in epididymal, mesenteric and subcutaneous adipose tissues (Fig. 3). Orlistat therapy did not cause any mentionable side-effects in the treated animals. No animals in any of the groups died during the study period.

Three different doses ($10^8$–$10^{10}$) of the three *L. sakei* strains (CJB38, CJB46 and CJLS03) were orally administered to high fat DIO C57BL/6 mice for 7 weeks, and body weight and food consumption were measured daily. During the test period, three strains were found to exhibit reduced weight gain compared to the HFD group (Figs. 1B–1D),

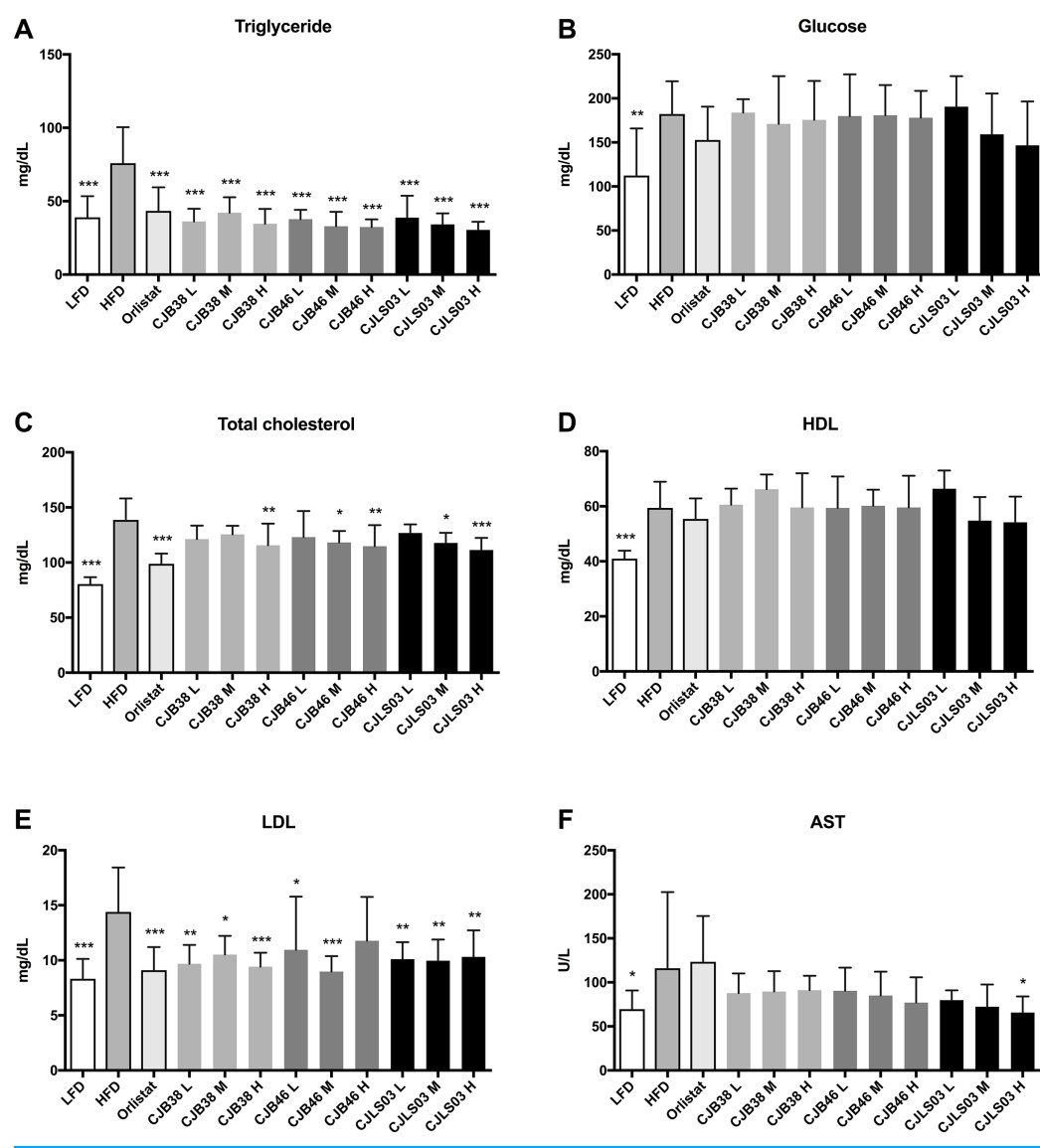

**Figure 2 Serum biomarkers of each experimental group showing (A) triglycerides, (B) glucose, (C) total cholesterol, (D) high density lipoprotein (HDL), (E) low density lipoprotein (LDL) and (F) aspartate transaminase (AST).** LFD, low-fat diet; HFD, high-fat diet; CJB38, CJB46 and CJLS03 denote the three *L. sakei* strains; the three dose levels of each strain administered together with the HFD were $1 \times 10^{10}$ CFU/mL (high-dose, H), $1 \times 10^{9}$ (medium-dose, M) and $1 \times 10^{8}$ CFU/mL (low-dose, L). The values for each index are expressed as the mean ± SD ($n = 10$). Asterisks denote the level of significance compared to HFD as $^{*}p < 0.05$, $^{**}p < 0.01$ and $^{***}p < 0.001$.

with strain CJLS03 showing, dose-dependently, the strongest effect of the three strains. LFD, Orlistat, the full CJB46 group, and medium and high dose of the CJLS03 groups showed significantly lower weight increase compared to the HFD group (Fig. 1E; Fig. S3). The weight loss of CJB38 or CJB46 was not dependent of the dose while only strain CJLS03 showed a dose-dependent weight reduction effect, and with the highest efficacy of all groups for CJLS03 H (Fig. 1E). The onset time of weight loss showed significance compared to the HFD at days 4, 21, 21 and 7 for the Orlistat, CJB38, CJB46 and

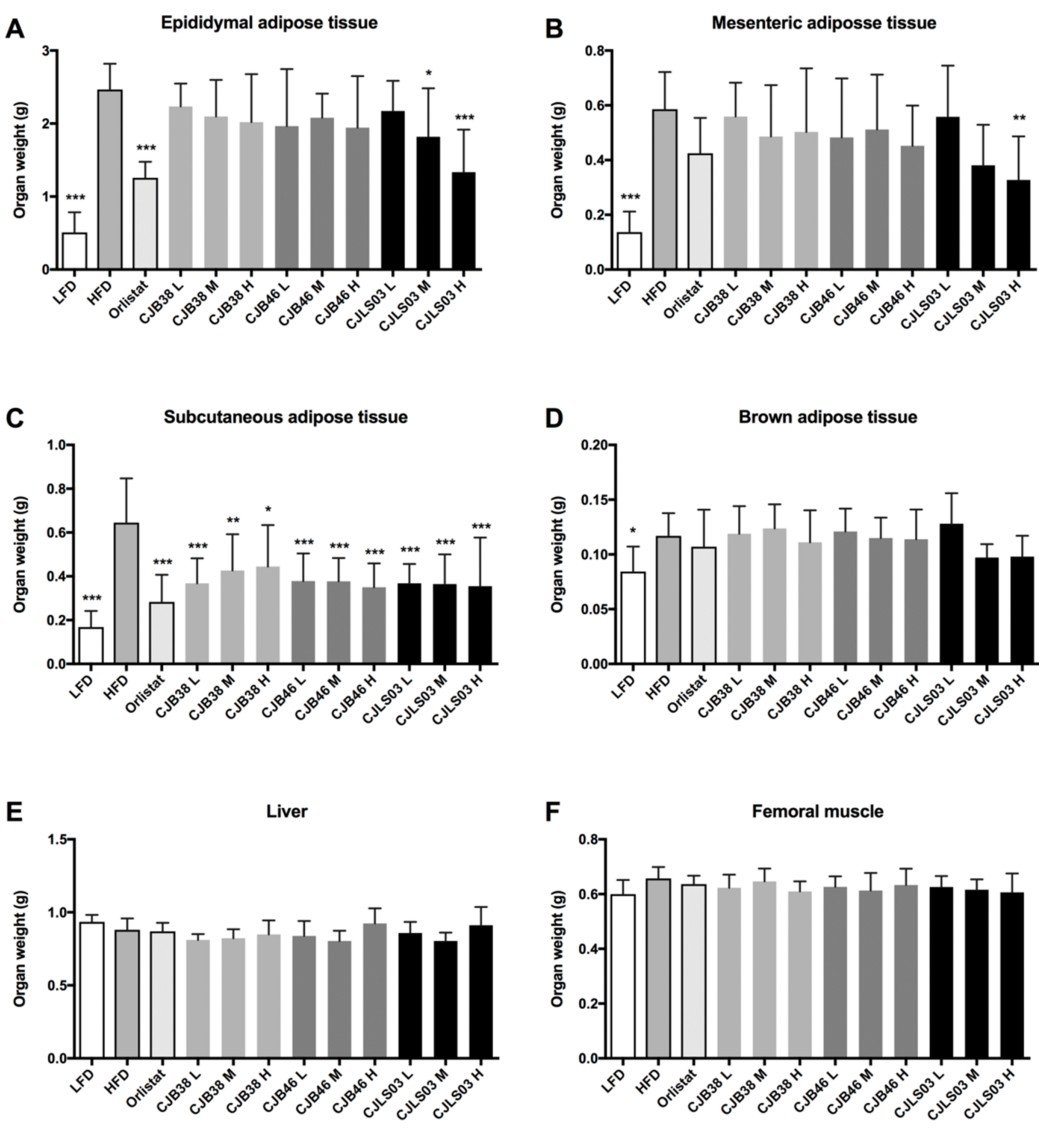

**Figure 3 Organ weights of each experimental group showing (A) epididymal adipose tissue, (B) mesenteric adipose tissue, (C) subcutaneous adipose tissue, (D) brown adipose tissue, (E) liver and (F) femoral muscle.** LFD, low-fat diet; HFD, high-fat diet; CJB38, CJB46 and CJLS03 denote the three *L. sakei* strains; the three dose levels of each strain administered together with the HFD were $1 \times 10^{10}$ CFU/mL (high-dose, H), $1 \times 10^9$ (medium-dose, M) and $1 \times 10^8$ CFU/mL (low-dose, L). The values for each index are expressed as the mean $\pm$ SD ($n = 10$). Asterisks denote the level of significance compared to HFD as $*p < 0.05$, $**p < 0.01$ and $***p < 0.001$.

CJLS03 groups, respectively (Table S2). The daily dietary intake was significantly higher in the LFD, Orlistat and CJLS03 M groups compared to the HFD group (Fig. 1F).

Serum biochemical analysis showed an overall increase in the lipid profile (TC, TG, HDL, LDL), liver (AST) and the GLU level of the HFD group compared to the LFD group, demonstrating that a HFD intake may impact various biomarkers associated with pathophysiological symptoms of obesity (Fig. 2). Compared to the HFD group, the serum TG level decreased in all test groups (Fig. 2A) while the LDL level was significantly reduced

in all test groups except CJB46 H (Fig. 2E). Significant reduction of TC was only detected in LFD, Orlistat and in the groups treated with higher doses (M and H) of *L. sakei* CJB38 H, CJB46 M, CJB46 H, CJLS03 M and CJLS03 H (Fig. 2C). In particular, the CJLS03 group, shown to be superior regarding weight gain inhibition, appears to be effective in a dose-dependent manner (Figs. 2A–2C). HDL levels were not significantly different from the HFD group in all the test groups, however, all *L. sakei* treated groups except CJB46 L, CJLS03 M and CJLS03 H showed significant increase when the ratio of HDL to TC level was calculated; this is reflected in Fig. 2D. Serum AST values (indicating liver function) were found to be approximately 1.7 times higher for the HFD compared to the LFD group (Fig. 2F), while the Orlistat group showed no significant change in AST level compared to the HFD group. All nine groups receiving the *L. sakei* strains showed a trend toward reduced AST levels but with only the high dose of CJLS03 (CJLS03 H) differing significantly when compared to the HFD group (Fig. 2F). CJLS03 showed the highest overall effectivity and a dose-dependent anti-obesity function; at the same time, it induced a dose-dependent improvement of serum obesity-associated biomarkers and liver function. Liver H&E staining optically demonstrated normal histology in LFD mice with minor lipid accumulation. Comparing the visual differences, the HFD-fed mice showed extensive fat accumulation and moderate vacuolations around the portal triad. In the groups treated with the higher dose of *L. sakei* CJB38 H, CJB46 H and CJLS03 H inhibition of lipid accumulation was visually evident and was comparable to that of the LFD group (Fig. S4).

Compared to HFD the LFD group showed significantly lower weights of epididymal, mesenteric, subcutaneous and brown adipose tissues while insignificant organ weight differences were measured in liver and femoral muscles (Fig. 3). Every dose of all three strains of *L. sakei* and the orlistat treatment resulted in significantly lower subcutaneous adipose tissue weight while only CJLS03 H showed significant reduction of visceral adipose tissue including epididymal and mesenteric adipose tissue, when compared to the HFD group (Figs. 3A–3C). CJLS03 M treatment significantly reduced epididymal adipose tissue weight when compared to the HFD group (Fig. 3A). These results suggest that the three different *L. sakei* strains inhibited the accumulation of subcutaneous adipose tissue but that the CJLS03 group responded by dose-dependent reduction of visceral adipose tissues including the epididymal and mesenteric adipose tissues (Figs. 3A and 3B). Orlistat and *L. sakei* treatment did not result in significant weight differences regarding brown adipose tissue, liver and femoral muscle (Figs. 3D–3F).

## DISCUSSION

The impact of a HFD on various biomarkers associated with pathophysiological symptoms of obesity is well established and supported in current literature (*Chandler et al., 2017*; *Lee, 2013*; *Ludwig et al., 2018*; *Siri-Tarino et al., 2010*). The body mass increase resulting from HFD feeding (as compared to a LFD) in this study (Fig. 1) was also accompanied by significant increases in serum biomarkers such as TG, TC, GLU, LDL and AST (Fig. 2) and also increases in epididymal, mesenteric and subcutaneous adipose tissues (Fig. 3). Definition of an ideal HFD and its exact composition is generally considered

difficult (*Buettner, Schölmerich & Bollheimer, 2007*). However, the standardization of the specific laboratory and feeding conditions for the purpose of metabolic studies is essential. In our studies, we have used exactly defined and commercially available HFD and LFD. The selected murine model (male C57BL/6 mice) is widely preferred as in vivo model for obesity and metabolic studies (*Khan et al., 2014*) and related investigations (*Neuhofer et al., 2014*).

The anti-obesity influence of administered probiotics is a heavily debated issue, yet, an indisputable fact is that the host gut microbiota is exercising a leverage over energy efficiency and adipose tissue accumulation (*Kobyliak et al., 2017*; *Greiner & Bäckhed, 2011*; *Delzenne et al., 2011*). At the same time, probiotics have been reported to impact the host microbiota in a positive way (*Hemarajata & Versalovic, 2013*) and to beneficially influence gut homeostasis and reduce the symptoms of gastrointestinal diseases (*Bron et al., 2017*). The beneficial effect of probiotics on the levels of alanine aminotransferase, AST, TC, HDL, tumor necrosis factor-α and also on insulin resistance (assessed in a homeostasis model (HOMA-IR)) have been reported earlier (*Ma et al., 2013*). In a study using C57BL/6J mice *L. rhamnosus* GG (LGG) showed a protective effect against nonalcoholic fatty liver disease (NAFLD) induced by a high-fructose diet (*Ritze et al., 2014*). This potential is supported by meta-analysis of data from randomized controlled trials in patients with NAFLD, showing probiotic therapy to result in a significant decrease of NAFLD (*Ma et al., 2013*; *Al-muzafar & Amin, 2017*). Moreover, probiotic therapy has been shown to be typically associated with a reduction in liver aminotransferase levels (*Aller et al., 2011*; *Buss et al., 2014*; *Shavakhi et al., 2013*). The significant reduction of liver AST levels by *L. sakei* CJLS03 H in our study suggests its possible therapeutic potential for alleviation of NAFLD. The potential advantages of probiotics as complementary treatment for metabolic disorders and as therapy for NAFLD are increasingly recognized (*Le Barz et al., 2015*; *Ma, Zhou & Li, 2017*). Moreover, the modulatory effect of probiotics on the gut microbiota suggests their potential as a "promising and innovative add-on therapeutic tool" for the treatment of NAFLD (*Paolella et al., 2014*). In our study, inhibition of hepatic lipid accumulation in HFD animals was revealed by Liver H&E staining and was particularly obvious for the groups treated with orlistat and CJLS03 H which also compared well with the normal histological features of the LFD group (Fig. S4).

The function of orlistat in assisting weight loss is well established and has been supported by Cochrane meta-analysis of various randomized controlled trials (*Drew, Diuxon & Dixon, 2007*). Obesity control may be by several mechanisms, one of which being that orlistat prevents fat hydrolysis by acting as a gastric and pancreatic lipase inhibitor (*Heck, Yanovski & Calis, 2012*; *Yanovski & Yanovski, 2014*). It has been successfully used as anti-obesity control in animal experiments involving high fat DIO rats (*Karimi et al., 2015*) and DIO C57BL/6 mice (*Chung et al., 2016*). The latter studies also included clinical trials, and the authors (*Chung et al., 2016*) claimed orlistat to be the most popular anti-obesity pharmaceutical drug, both in animal (DIO C57BL/6 mice) experiments and clinical trials. The DIO C57BL/6 mouse is now widely accepted as an in vivo model of choice. It has been reported to closely reflect human metabolic disorders such as obesity, hyperinsulinemia, hyperglycemia and hypertension (*Collins et al., 2004*).

In particular, the metabolic abnormalities of DIO C57BL/6 after HFD feeding are considered reported to closely resemble those of human obesity development patterns (*Speakman et al., 2007*), and also regarding properties such as adipocyte hyperplasia, fat deposition in the mesentery and increased fat mass (*Inui, 2003*).

Probiotic administration increasingly enjoys consideration as a promising approach for beneficially modulating the host microbiota (*Jia et al., 2008*; *Steer et al., 2000*). Numerous reports confirmed the beneficial effects of specific probiotic strains against diarrhoea and inflammatory bowel diseases (*Ahmadi, Alizadeh-Navaei & Rezai, 2015*; *Gionchetti et al., 2000*; *Ouwehand, Salminen & Isolauri, 2002*). Recently, anti-obesity effects of probiotics were also reported and confirmed in clinical trials (*Kadooka et al., 2010*; *Woodard et al., 2009*; *Minami et al., 2015*, *2018*; *Borgeraas et al., 2017*) and animal models (*Kim et al., 2016*; *Alard et al., 2016*; *Wang et al., 2015*; *Ji et al., 2012*). *Kadooka et al. (2010)* investigated the anti-obesity effect of the probiotic *L. gasseri* SBT2055 by conducting a double-blind, randomized, placebo-controlled intervention trial with 87 overweight and obese subjects for 12 weeks. The data confirmed that the abdominal visceral and subcutaneous fat area, weight, BMI, as well as waist and hip measures were significantly reduced in the group consuming the probiotic. In another study (*Woodard et al., 2009*) 44 morbid obese patients were operated for weight loss by surgery (gastric bypass surgery) and were randomly divided in a probiotic administered group and a control group. A significantly higher weight loss was recorded in the group receiving the probiotic (described as "Puritan's Pride®," containing a mixture of 2.4 billion live cells of *Lactobacillus* spp.). *Park et al. (2013)* reported a significant weight reduction of a C57BL/6 mice model after *L. curvatus* HY7601 and *L. plantarum* KY1032 consumption, however, faecal microbiota modulation of major groups such as *Firmicutes* and *Bacteroidetes* was not monitored.

One of the major hurdles for an accurate clinical trial is to understand the effective dose of a probiotic at a strain-specific level. Selecting the correct dose of a probiotic for a specific purpose such as the alleviation of diarrhoea was suggested in various studies; yet, there is a general lack of scientific proof of a concept to define the functional dose of a probiotic (*Kollaritsch et al., 1989*, *1993*; *Islam, 2016*). *Chen et al. (2015)* used a range of five different 10-fold doses of *L. acidophilus* in a colitis-induced animal model and reported $10^6$ CFU/10 g of the animal weight as the most effective application level for modulating the bacterial profile in the distal colon. In our study we have monitored dose-related effects of three different strains of *L. sakei* and found only one strain, CJLS03, to show a dose-dependent anti-obesity effect while the anti-obesity impact of the other two strains was lower and dose-independent (Fig. S3). At dose levels from $1 \times 10^8$ to $1 \times 10^{10}$ CFU/mL administration of strain CJLS03 resulted in a dose-related (progressive) reduction in the levels of TC, TG, AST, mesenteric adipose tissue and epididymal adipose tissue (Fig. S3). Adipose tissues were reduced relative to weight gain, and TG and TC showed the most significant reduction in the *L. sakei* treated groups compared to the HFD control group. Another *L. sakei* strain (OK67) isolated from kimchi was reported to ameliorate HFD-induced blood GLU intolerance and obesity in mice; mechanisms for this effect have been suggested to be by inhibition of gut microbial

lipopolysaccharide production and the inducing of colon tight junction protein expression (*Lim et al., 2016*).

Our study has confirmed the relevance of a strain-specific approach when selecting functional strains suitable for (costly and time-consuming) clinical studies. The importance of this issue has been emphasized in recent papers with regard to pre-clinical physiological studies on putative probiotic strains of lactic acid bacteria and *Bifidobacterium*. These studies involved features such as adhesion potential, antibiotic resistance and survival under simulated conditions of the upper GIT, in addition to the modulation of the gut microbiome (*Bubnov et al., 2018*).

## CONCLUSIONS

This in vivo investigation showed that beneficial effects of putative probiotics are both strain-specific and dose-related. For only one (CJLS03) out of three *L. sakei* strains an anti-obesity effect could be detected, which, at the same time, was found to be dose-dependent. The highest of three doses ($1 \times 10^{10}$ CFU/day) of CJLS03 gave the most favorable (significant) biomarker-related effects with regard to cholesterol and triglyceride reduction, when compared to the HFD control.

### Funding

This work was supported by the CJ CheilJedang Corporation, Seoul, South Korea, and the Bio and Medical Technology Development Program of the National Research Foundation (NRF) No. 2016M2A9A5923160 and 2018M3A9F3021964 (Ministry of Science, ICT & Future Planning). There was no additional external funding received for this study. The funders had no role in study design, data collection and analysis, decision to publish, or preparation of the manuscript.

### Grant Disclosures

The following grant information was disclosed by the authors:
Bio and Medical Technology Development Program of the National Research Foundation (NRF): 2016M2A9A5923160 and 2018M3A9F3021964.

### Competing Interests

Yosep Ji, Soyoung Park and Wilhelm H Holzapfel have received research grants, via Handong Global University, from CJ CheilJedang Corporation, South Korea. Co-authors Young Mee Chung, Dahye Jeong and Bongjoon Kim are employed by CJ CheilJedang Corp., Blossom Park, Republic of Korea.

### Author Contributions

- Yosep Ji conceived and designed the experiments, performed the experiments, analyzed the data, contributed reagents/materials/analysis tools, prepared figures and/or tables, authored or reviewed drafts of the paper, approved the final draft.

- Young Mee Chung performed the experiments, analyzed the data, prepared figures and/or tables, authored or reviewed drafts of the paper, approved the final draft.
- Soyoung Park performed the experiments, analyzed the data, prepared figures and/or tables, authored or reviewed drafts of the paper, approved the final draft.
- Dahye Jeong conceived and designed the experiments, contributed reagents/materials/analysis tools.
- Bongjoon Kim conceived and designed the experiments, contributed reagents/materials/analysis tools, authored or reviewed drafts of the paper.
- Wilhelm Heinrich Holzapfel conceived and designed the experiments, contributed reagents/materials/analysis tools, authored or reviewed drafts of the paper, approved the final draft.

## Animal Ethics

The following information was supplied relating to ethical approvals (i.e., approving body and any reference numbers):

The animal study was approved by the Ethical Committee of KPC Ltd. in Korea (P150067) in full compliance with ethical standards as specified by Korean law. KPC Ltd. is a commercial research institution dealing with contracted animals studies, and fully complies complying with Government standards for conducting animal studies. These include the involvement of a medical doctor and/or a veterinarian.

## Data Availability

The raw data are available in the Supplemental Files.

## Supplemental Information

Supplemental information for this article can be found online at http://dx.doi.org/10.7717/peerj.6651#supplemental-information.

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
