# Peer review of "Dose-dependent and strain-dependent anti-obesity effects of Lactobacillus sakei in a diet induced obese murine model"

_PeerJ, doi:10.7717/peerj.6651_

## Round 0.1 · original submission · Major Revisions

Apologies for the delay in informing you of the editorial decision on your article. Work commitments have been heavy the past couple of weeks.

You will see that your article has been reviewed by two experts, who have highlighted areas that you need to address during revision.

I agree with one of the reviewers with respect to the Methods. Additional information is needed in a number of instances.
- Provide composition information for both diets.
- 120 animals were split into 12 treatment groups. How many animals per cage? How were potential cage effects addressed?
- It is stated that "each group was treated" with bacteria, orlistat or control. What this by oral gavage or some other means? This information must be included in the text. (I note that oral gavage is mentioned later for the L. sakei strains, but it needs to be made clear if this also applies to the orlistat control and negative control).
- What other rodent studies have used orlistat as a chemical control for its anti-obesity effect?
- Side-effects experienced by animals given orlistat therapy should be reported in the article. Did any side-effects persist, worsen or lessen over the 7 weeks of the study? Indicate whether the orlistat was insoluble in PBS.
- Indicate if any animals died during the study. If so, how many?
- On what basis were the three tested L. sakei strains selected? Relevant information/literature relating to the isolation/characterization (taxonomic, bile acid tolerance, pH tolerance) of the strains should be included.
- Provide the equation and the pre-optmised standard curves for the three L. sakei strains as supplementary material.

It is unsurprising that HFD increased AST levels in liver. Non-alcoholic fatty liver disease is the hepatic manifestation of metabolic syndrome and associated with HFD. Appropriate citations and comment in relation to this aspect of metabolic disease should be included in the article. Especially as CJLS03 H appears to have significantly reduced liver AST levels, and studies have shown therapeutic potential of probiotics to alleviate aspects of NAFLD (e.g. PMID: 30113661). If you have information on level of steatosis (e.g. via oil red O staining) and/or liver triglycerides for the hepatic tissue, this should be included in the article and discussed in the context of hepatic steatosis.

Why are no statistical results reported for Fig. 1b-d?
You write "LFD, Orlistat, all of the CJB46 group, and medium and high dose of the CJLS03 groups showed significantly lower weight increase compared to the HFD group." Are you referring to Fig. 1e here or 1b-d? Make clear.

"The onset time of weight loss showed significance compared to HFD at days 4, 21, 21 and 7 for the Orlistat, CJB38, CJB46 and CJLS03 groups, respectively." Include these data in supplementary material.

"high fat diet intake may impact various biomarkers associated with
pathophysiological symptoms of obesity". This is well established in the literature, and should be acknowledged in the current article by reference to appropriate publications.

Figures 1-3. For each panel in each figure, numbers of animals per group should be reported.

Reviewer 1 ·

Basic reporting

Review on PeerJ # 29824
« Dose-dependent anti-obesity effects of three different Lactobacillus sakei strains using a diet induced obese murine model”
General comments:
Yosep Ji and co-workers present an article on the anti-obesity effect of 3 strains of Lactobacillus sakei. In the context of growing incidence of metabolic syndrome, obesity, and associated morbidity and mortality, the rational of this research is obvious. In this context, attention should be paid to the microbial part of our diet, and this is mainly provided in fermented foods. It is thus wise to seek modulation of obesity-induced pathology as a result of L. sakei consumption, as it is a member of the Kimchi microbiota. Interestingly, the authors address the dose-dependence and the strain-dependence of anti-obesity effects in of L. sakei. Indeed, two strains, within the same bacterial species, may have completely different properties so that one cannot conclude on the beneficial attributes of a species, based on a single study. The paper is technically sound. The manuscript is clearly written. The reviwer recommends publication of this paper, provided that minor modifications are done.
Minor modifications:
The reviewer recommends changing the title: “Dose-dependent and strain-dependent anti-obesity effects Lactobacillus sakei revealed using a diet-induced obese murine model”
Dose-dependent and strain-dependent, with an hyphen: correct throughout the paper
Line 30&31: rephrase, it is not clear: cause of various…..syndrome
Line 31: pathogenicity: of what?
Line 32: what does remains a poorly defined area?
Line 38: “various … trials”: then more clinical studies should be cited in the paper
Line 66&67: rephrase. Deeper understanding of pathogenesis is needed…..knowledge…is limited
Line 70 and elsewhere : the microbiota
71&72: new clinical diagnostic perspective: define, explain
82: (including development of dedicated in vitro models)
84: distinguish cited preclinical and cited clinical studies
88 to 90: cut the sentence
100 to 109: please evidence the link between these cited reports and your present work.
110 to 112: cut the sentence
117&118: composition of the regimen could be given in a table
128: what was the dose of Orlistat per day? Describe briefly the effect of Orlistat. You could also cite a reference reporting its use in a similar animal study.
134: AST: describe, a marker of ……153: you should add a sentence like “at the end of the treatment, body weight was increased by the HFD. This weight gain was limited by…(Fig.1A)….This figure is not cited…..
RESULTS: use capital letters: 1A, 1B, ….
151-161: it should be mentioned here that only strain CJLS03 has a significant and important effect, I think.
177: noticeable decrease…..results being non-significant, I suggest “a trend in AST reduction”
185: Orlistat treatment resulted in….
208: some selected strains of probiotics have been….208-209: animal studies (…) and clinical studies (…) : separate the citations
209: clinical studies : you should indicate here Kadooka and Woodard, you should also cite the work of Minami et al. : J Nutr Sci. 2015 and Biosci Microbiota Food Health. 2018.
217: the probiotic: which one?
218: in a mice model, as a result of L. curvatus….consumption231: …was found to be lower and dose-independent.

Experimental design

Experimental design is correct, authors could better described the origin of the strains, their use in previous studies, the composition of the diets, LF, HF.

Validity of the findings

Data is robust

·

Basic reporting

I have read manuscript titled `Dose dependent anti-obesity effect of three different Lactobacillus sakei strains using a diet induced obese murine model` by Yosep Ji et al.
Topic is very interesting and of high relevance for appropriate use of probiotics.
However, some improvements can be done / considered to be implemented into current version.
Thus, some relevant aspects seem missed.
My first concern is is relevance of the model and validity of data supporting dose dependant effect.
Introduction should include review of evidence from recent sources in respect to effects of (1) Probiotics – (2) Lactobacillus strains – (3) Lactobacillus sakei on metabolic syndrome cholesterol levels, on type 2 diabetes mellitus, etc., every point in particular;
and refer to most appropriate papers in the field(s), like:
[Hill C, Guarner F, Reid G, Gibson GR, Merenstein DJ, Pot B, et al. Expert consensus document. The International Scientific Association for Probiotics and Prebiotics consensus statement on the scope and appropriate use of the term probiotic. Nat Rev Gastroenterol Hepatol. 2014(8):506–14. https://doi.org/10.1038/nrgastro.2014.66
Reid G, Abrahamsson T, Bailey M, et al. How do probiotics and prebiotics function at distant sites? Benef Microbes. 2017 Jul 20:1-14. DOI 10.3920/BM2016.0222
Bubnov RV, Babenko LP, Lazarenko LM, Mokrozub VV, Demchenko OA, Nechypurenko OV, Spivak MY. Comparative study of probiotic effects of Lactobacillus and Bifidobacteria strains on cholesterol levels, liver morphology and the gut microbiota ino bese mice. EPMA J. 2017 Oct 10;8(4):357-376. doi: 10.1007/s13167-017-0117-3.
Etc.].
This should suggest the hypothesis and the novelty.
Strain / dose specific properties can be considered in – see [Specific properties of probiotic strains: relevance and benefits for the host EPMA J 2018. DOI : 10.1007/s13167-018-0132-z].

Experimental design

Materials section should be written in more detailed manner. Authors can descibe the model, it is not clear in current version, provide the scheme of experiment.
When the probiotics was given? What was content of HFD?
Why orlistat was used? Does/might it have beneficial effects on metabolic profile??
Was the microbiota analyzed?
What was the duration and stability of effect (dose dependant)?
Some of Matmet data are presented in the Results section.
Some point might be discussed like:
• What conclusions can be done based on differences found in brown adipose tissue, liver and femoral muscle?
• Association of microbiome with obesity and mechanics of beneficial effects and potential implementation to clinical practice.
• Is L. sakei specific strain for obesity? Is it a part of fermented food?
• How results can highlight new treatment / prevention approaches in clinical set?
Clear and concise endnotes can be added in the end of the paper.

Validity of the findings

Author studied strain Lactobacillus sakei administered in a high fat diet induced obese murine model using three different doses, 1x1010 CFU, 1x109 CFU and 1x108 CFU.
Authors`s results show that only one strain of L. sakei (CJLS03) induced a dose dependent anti-obesity effect.
Statistical significance is clear, however, the relevance of the model used is of high importance.

Additional comments

I encourage Authors to reconsider article in order to clarify points mentioned above.

---

## Round 0.2 · Minor Revisions

Thank you for taking on board the reviewers' comments and submitting a significantly improved article. There are still a few minor issues I would like you to deal with before your article is ready for publication. Please refer to the reviewer's comments and address all. Please also find attached the Word file of your revised manuscript with some minor typographical issues and comments that need to be dealt with. I look forward to receiving your revised work in due course.

·

Basic reporting

On my opinion, Authors have implemented extensive improvement into the manuscript and responsed to reviewers` comments. All corrections are relevant.
The manuscript is about to ready for acceptance.
And minor points should be considered for correction before publication, like:

In the Abstract: Conclusion vs `Discussion. This study shows intraspecies …`
I have missed in the text clear description of components of low-fat (LFD) and high-fat (HFD) diets used in this study, only the chemical compounds in Table 2.
Has been the low fat or rather low industrial fat diet considered in research group? Names of products might help understanding and generate dietary recommendations.

Limitation paragraph can be included.

Finallly, once again careful checking for spelling needed.

Experimental design

In the revised version with explanations done the design is relevant to the studied topic.
Methods described with sufficient detail & information to replicate.
however, I have missed in the text clear description of components of low-fat (LFD) and high-fat (HFD) diets used in this study, only the chemical compounds in Table 2.
Has been the low fat or rather low industrial fat diet considered in research group? Names of products might help understanding and generate dietary recommendations.

Validity of the findings

The findings and conclusions of the study are valid, however limitation paragraph can be included.

Additional comments

On my opinion, Authors have done extensive improvement of the manuscript and responsed to reviewers` comments. All corrections are relevant.
The manuscript is about to ready for acceptance,
minor points can be suggested for correction before publication.

---

## Round 0.3 · accepted · Accept

Thank you for addressing the last few minor points of the reviewer and for making recommended typographical changes. I am happy to accept your article for publication.

#